# A Vision Servo System for Automated Harvest of Sweet Pepper in Korean Greenhouse Environment

**BongKi Lee [1], DongHwan Kam [1], ByeongRo Min [1,*], JiHo Hwa [2] and SeBu Oh [3,*]**

[1] Institute of Biotechnology and Bioengineering, Sungkyunkwan University, Suwon-si 16419, Korea; dkways@skku.edu (B.L.); kamdh@skku.edu (D.K.)
[2] Manufacturing Technology Center, LS Mtron, Gunpo-si 15845, Korea; Youniri@nate.com
[3] Department of Mechanical Engineering, Chungang University, Seoul-si 06974, Korea
[*] Correspondence: min7887@skku.edu (B.M.); ok777@cau.ac.kr (S.O.);
Tel.: +82-31-353-6152 (B.M.); +82-2-820-5314 (S.O.)

**Abstract:** Recently, farmers of sweet pepper suffer from the increase of its unit production costs due to the rise of labor costs. The rise of unit production costs of sweet pepper, on the other hand, decreases its productivity and causes the lack of its farming expertise, thus resulting in the quality degradation of products. In this regard, it is necessary to introduce an automated robot harvest system into the farming of sweet pepper. In this study, the authors developed an image-based closed-loop control system (a vision servo system) and an automated sweet pepper harvesting robot system and then carried out experiments to verify its efficiency. The working area of the manipulator that detects products through an imaging sensor in the farming environment of sweet pepper, decides whether to harvest it or not, and then informs the location of the product to the control center, which is set up at the distance scope of 350~600 mm from the center of the system and 1000 mm vertically. In order to confirm the performance of the sweet pepper recognition in this study, 269 sweet pepper images were used to extract fruits. Of 269 sweet pepper images, 82.16% were recognized successfully. The harvesting experiment of the system developed in this study was carried out with 100 sweet peppers. The result of experiment with 100 sweet peppers presents the fact that its approach rate to peduncle is about 86.7%, and via four sessions of repetitive harvest experiment it achieves a maximal 70% harvest rate, and its average time of harvest is 51.1 s.

**Keywords:** sweet pepper harvesting automation; cylindrical robot; vision servo system; image processing-based; momentum backpropagation

---

## 1. Introduction

Manpower decrease and aging in the agricultural sector for food production cause the price rise of farm products. In order to keep up farmer's productivity and competitiveness, therefore, it is necessary to improve and automate agricultural technology [1].

More than 20,000 tons of sweet pepper (in other words, bell-pepper), one of South Korea's most cultivated greenhouse plants, are exported abroad annually, with the share of South Korean sweet pepper in the Japanese market exceeding 60% [2,3]. However, with the recent increase of its unit production cost due to the continuous rise of labor costs, the productivity of sweet pepper is decreasing continuously in South Korea [4]. In addition, the commercial value of sweet pepper is reduced by the harvesting of day laborers lacking harvesting expertise [5]. In order to solve these problems of sweet pepper harvesting in South Korea, it is necessary to introduce a robot-based automated harvesting system. In order to carry out a robot-used automated harvest of sweet pepper, it is necessary to introduce a system which can consider the farming environment of the greenhouse and the location,

size, shape and color of fruits. It is also necessary to reduce the problem of damaging the stem of sweet pepper in the course of the harvest.

In the cases of South Korea, there has been active research on the automation of pest control and the development of automation technology for agricultural work in the interest of crops, but little research on fruit recognition and harvesting has been conducted recently [6]. In South Korea, Ha and Kim [7] developed a harvesting robot, which was designed to be suitable for harvesting oriental melon in a greenhouse, using a 4-axis manipulator structure combining shuttle-type and orthogonal coordinate-type. Hwang et al. [8] developed multi-functional robot which was designed to cultivate watermelon in a South Korean greenhouse. Min et al. [9,10] in South Korea developed an end-effector and a manipulator for a cucumber harvest robot. In contrast to South Korea, researchers around the world are actively studying harvest robots to automate harvesting operations. Fruit harvesting robots and fruit recognition methods have been actively studied for the automation of harvesting work [11]. Lin et al. [12] studied detection methods of guava fruit using an RGB (red, green, blue) depth camera, and then used a state-of-the-art FCN (Fully Convolutional Network) model [13] to segment the guava fruits and branches from aligned RGB images obtained using the RGB depth camera. The precision and recall of guava fruit detection were 0.983 and 0.948, respectively. Ostovar et al. [14] studied image processing methods to detect yellow peppers for a harvest robot and suggested a method to threshold yellow pepper fruits on a real greenhouse image. A decaying epsilon-greedy algorithm was proposed to threshold the fruit area of yellow peppers, and the performance of the algorithm to threshold the yellow pepper fruit area was 91.5%. Wang et al. [15] suggested the stereo matching method to harvest litchi fruits. Binocular CCD (Charge-Coupled Device) cameras were used to calculate the coordinates of litchi fruits in real-world coordinates; the highest matching success rate was 97.37% and the lowest matching success rate was 91.96%. Sa et al. [16] studied the detection method of sweet pepper peduncle for a harvest robot. RGB depth camera was used to detect peduncles of sweet pepper, and color and geometry information acquired from the RGB depth camera were used, and then a supervised-learning approach for the peduncle detection task was utilized; it achieved an AUC (the area under the curve) of 0.71 regarding detection precision of peduncles on field-grown sweet peppers. Kitamura and Oka [17] suggested a method to recognize sweet pepper fruits for a picking robot in a greenhouse. Regions of sweet pepper fruit were detected using the reflection characteristic (that fruits reflected light more than leaves) with LED (Light Emitting Diode) lights, and the recognition success rate for detecting sweet pepper fruits was 79.2%. Xiong et al. [18] developed a harvesting robot with a cable-driven gripper to harvest strawberries. A cable-driven gripper used in the system was designed to be robust against local errors introduced by the vision module, and the harvesting success rate was 53.57%. Barth et al. [19] suggested a method of angle estimation between the fruit and stem for a harvest robot. The method used to estimate the angle between the fruit and stem theoretically increased the success rate from 14% to 52% and impacted the harvest performance.

In order to automate the harvest work, a suitable harvest algorithm is needed for each country's cultivation environment. Compared to other countries, South Korea lacks research on the development of harvest robots for the automation of agricultural work [6]. This study introduces a vision servo control (image-based closed-loop control) [20] system for a Korean greenhouse environment to improve the accuracy of sweet pepper harvest. In addition, this study developed an automated sweet pepper harvest robot system based on a manipulator and an end-effector which could detect a sweet pepper via stereo cameras, decide whether or not to harvest it, carry out the work of harvest using the acquired information, and then verify its efficiency.

## 2. Material and Methods

### 2.1. Hardware Composition

As shown in Figure 1, the manipulator used in this study is the cylindrical robot which has three axes, namely a rotation axis, a horizontal axis, and a vertical axis, and is appropriate for a narrow space.

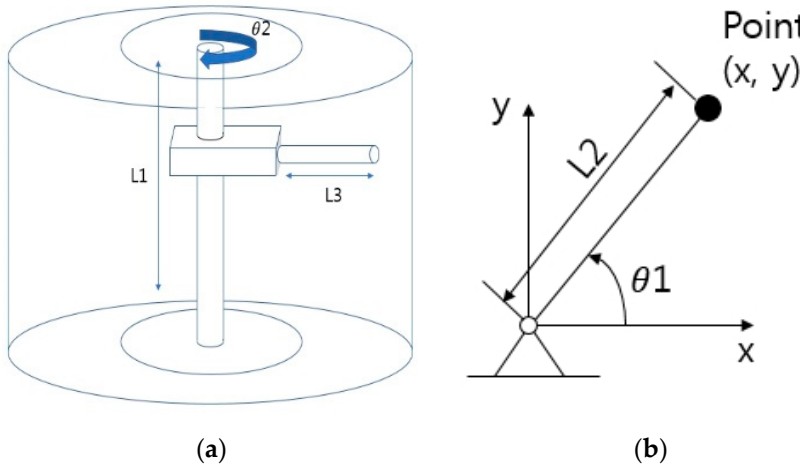

**Figure 1.** Coordinate relationship of the manipulator used in this study: (**a**) three axes (rotation, horizontal, vertical); (**b**) two axes ignoring vertical orientation.

The cylindrical robot is controlled by height L1, angle θ2, and radius L3 like Figure 1a. The robot can carry out works in a narrow space, like a sweet pepper greenhouse, efficiently and easily move to horizontally. It has three independent degrees of freedom, and its vertical and horizontal joints cross each other. If the degree of freedom of horizontal orientation is ignored, it is a rotating and horizontally moving L2-Θ2 manipulator as shown in Figure 1b. The rotation angle θ1 is defined as horizontal X-axis, and L2 can be defined as the distance from the rotation axis to the endpoint of the end-effector.

Depending on the environmental characteristics of the sweet pepper farming analyzed by Yu et al. [21] and Myung [22], the working area was set up at 350~600 mm from the center of the system and 1000 mm vertically. The planting interval of sweet pepper was 1000 mm and the rail intervals were 500~600 mm. The working area was set up after considering these conditions. In order for the manipulator to be able to transport horizontally and vertically two linear actuators were used. As for the linear actuator, which is in charge of vertical transportation: its stroke is 1000 mm; the lead of its ball thread is 10 mm; its maximal vertical load(weight) is 18 kg; its maximal transportation speed is 500 mm·s$^{-1}$; and its repetitive transfer accuracy is 0.005 mm. As for the linear actuator which oversees horizontal transportation: its stroke is 500 mm; its maximal load (weight) is 25 kg; and other specifications are the same with the linear actuator which oversees vertical transportation. The cylindrical robot is made of duralumin, an alloy of aluminum with excellent corrosion resistance, in consideration of a greenhouse's hot and humid working environment.

Figure 2 presents the manipulator with three axes, namely a rotation axis, a vertical axis, and a horizontal axis. Since the motor of the manipulator must work under the high temperature and humidity of the planting environment of sweet pepper, this study used Panasonic Minas a5 (Panasonic, Kadoma, Japan).

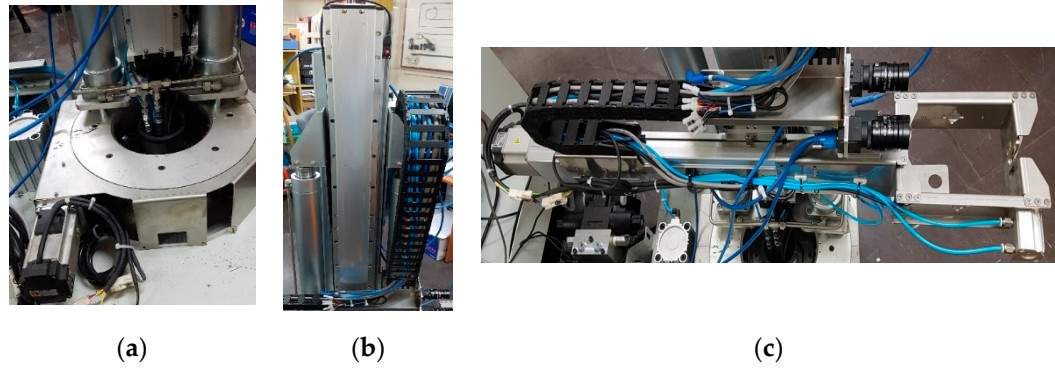

**Figure 2.** Manipulator: (**a**) rotation axis; (**b**) vertical axis; (**c**) horizontal z axis.

Figure 3 presents the end-effector. The end-effector props a fruit from below and makes the peduncle of the fruit positioned in its cutting portion through pose control, thus making its pneumatic cylinder harvest the fruit. The cutting part of the end-effector is designed to be carried out vertically, thus preventing damage to the stem of the sweet pepper. The end-effector was designed according to the characteristics of sweet peppers cultivated in South Korea with reference to the studies of Kim et al. [23], Ministry of Agriculture, Food and Rural Affairs (MAFRA) [24] and Um et al. [25]. The horizontal length of the end-effector is set up at 120 mm, and 150% of the sweet pepper's average diameter 80.92 mm. On the other hand, its vertical length is set up at 140 mm, and 150% of the sweet pepper's average length 92.15 mm. The entrance size of the cutting portion is set at 16 mm, in consideration of the average diameter of a sweet pepper's peduncle, thus preventing the peduncle's escape from the entrance. The entrance is set up at 10 mm initially, at a relatively narrower size, thus making it controlled elasticity by a spring. In the front of the end-effector, a pneumatic cylinder for cutting was installed, and in order to catch the stem of sweet pepper, a guide of 35 mm thickness is designed. For the stem of sweet pepper to be located in the guide, the guide has a 60° angle. In addition, the vertical guide is designed to be 59°. This is because the angle from the entrance of the guide to the portion of cutting is about 59° when the peduncle is assumed to be in the center of a sweet pepper fruit.

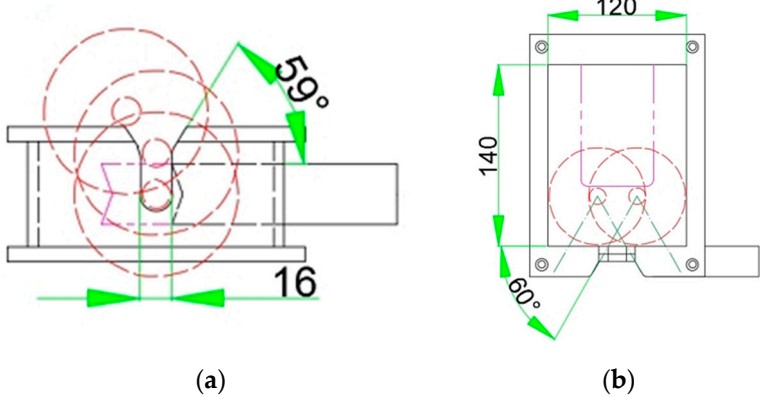

**Figure 3.** Design of end-effector: (**a**) frontal view; (**b**) plane view (black lines: end-effector contour, red lines: examples of sweet pepper's fruit).

Figure 4 presents the end-effector produced according to its design. At the bottom of the end-effector, a prop is installed in order to support sweet pepper fruits throughout the process from harvesting to transportation.

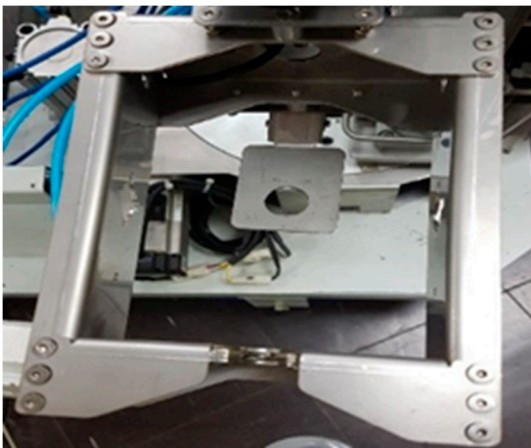

**Figure 4.** End-effector.

In this study, as shown in Figure 5, three cameras were used to acquire image information of sweet peppers in the greenhouse. The BFLY-U3-13S2C-CS model (FLIR, Portland, USA) of cameras was used. Cam1 and Cam2 were used as a stereo vision system to measure the distance to the sweet peppers. Cam3 was installed on the end-effector and used to correct the pose of the end-effector. To control the vision servo system, I5-4690 (Intel, Santa Clara, USA, 2017) and 8GB of Ram were used, and the vision servo system was programmed using C++ language on Windows 7 64bit OS (Microsoft, Redmond, USA, 2009) and Visual studio 2017 (Microsoft, Redmond, USA, 2017).

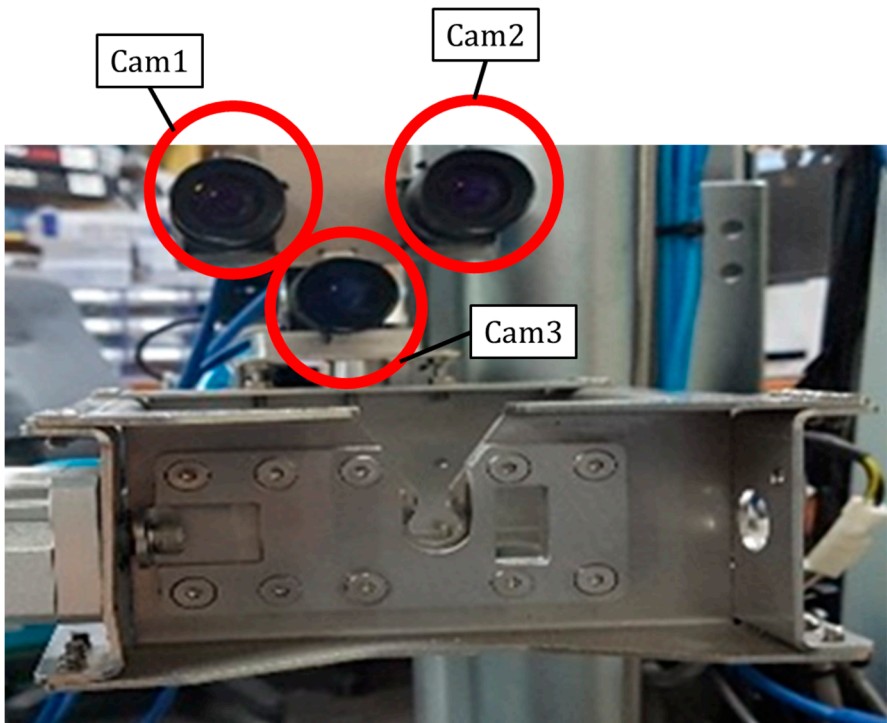

**Figure 5.** Three cameras used in the vision servo system.

Figure 6 shows images of artificially characterizing points of fruit of a red pepper using a laser light source. In this system, a green laser light source was used to easily extract feature points from the internal region of the fruit of the pepper. Figure 7 shows the stabilization of the laser light source designed for the manipulator and camera system. It is designed so that it does not interfere with other

systems when driving, and it is designed to install three laser light sources. The laser source was attached between Cam1 and Cam2.

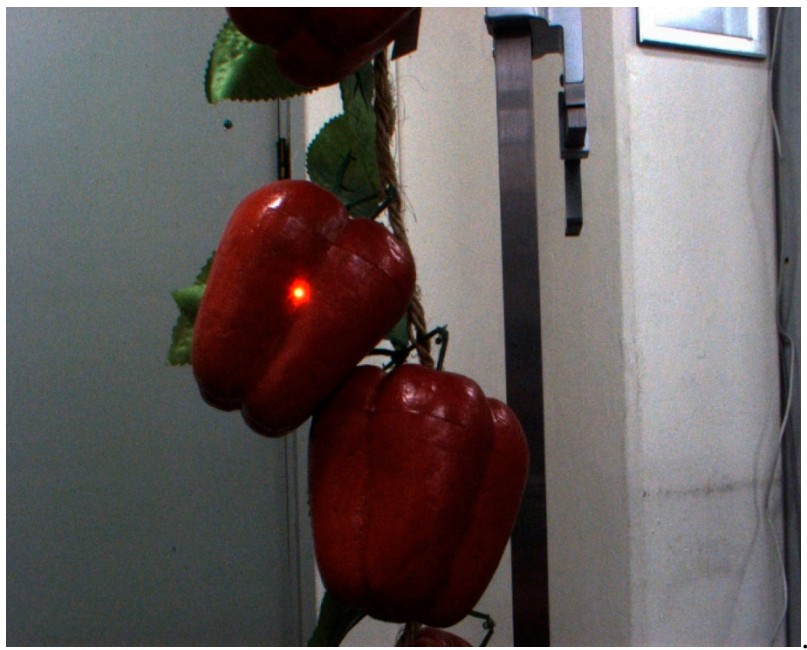

**Figure 6.** Illuminations using laser source.

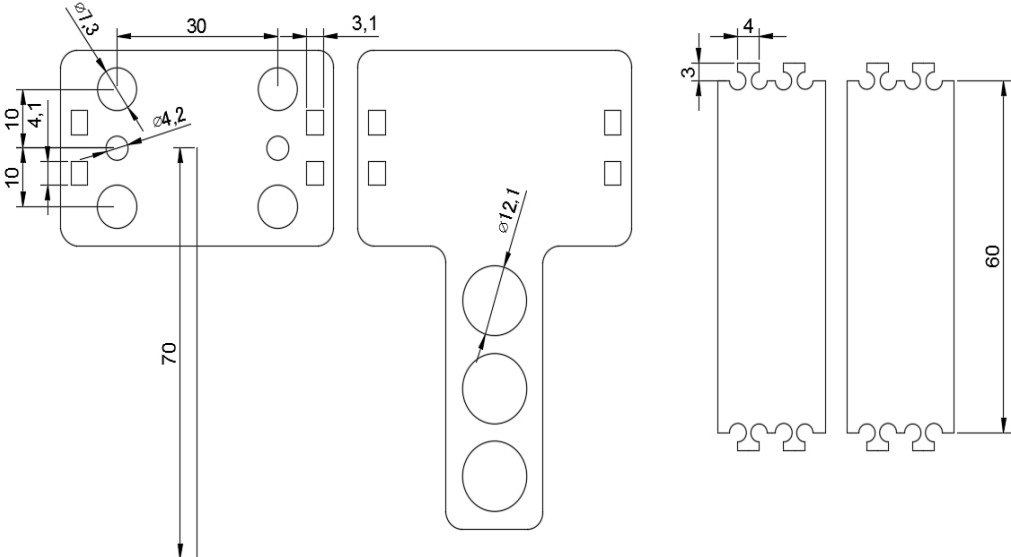

**Figure 7.** A design of joint parts for laser sources.

Figure 8 shows the stereo vision system (Cam1 and Cam2) attached laser source and shows the end-effector attached to Cam3. The system of this study was designed to control the end-effector to match the direction of the sweet pepper after extracting the center of the sweet pepper through Cam3 attached to the end-effector. Using the laser source attached between Cam1 and cam2, artificially characterizing points were created in the sweet pepper, and feature points extraction and stereo matching were performed through Cam1 and Cam2 to measure the distances.

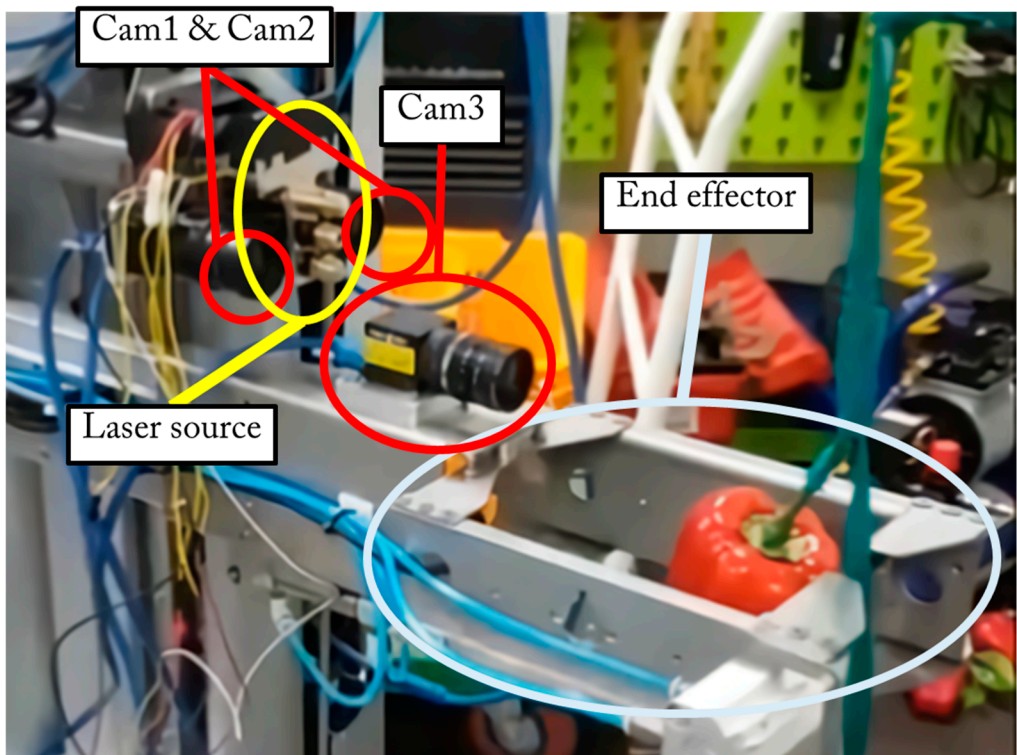

**Figure 8.** The stereo vision system (Cam1 and Cam2) attached laser source and end-effector attached to Cam3.

### 2.2. Sweet Pepper Recognition

Barnea et al. [26] used color information to recognize sweet pepper fruits. The color information of RGB (red, green, blue), hue, saturation, Y, Cb, and Cr in the fruit region of sweet peppers were used as learning factors of the momentum backpropagation algorithm. In general, images are represented by an RGB color model, and hue, saturation, Y, Cb, and Cr are represented by converting the RGB information of these images.

$$Hue = \begin{cases} \theta & if\ B \leq G \\ 360 - \theta & if\ B > G \end{cases}, \tag{1}$$

$$Saturation = 1 - \frac{3}{R+G+B}[\min(R,G,B)], \tag{2}$$

$$\theta = \cos^{-1}\left(\frac{\frac{1}{2}[(R-G)+(R-B)]}{\sqrt{(R-G)^2 + (R-B)(G-B)}}\right), \tag{3}$$

$$\begin{aligned} Y &= 0.257R + 0.504G + 0.095B + 16 \\ Cb &= -0.148R - 0.291G + 0.439B + 128, \\ Cr &= 0.439R - 0.368G - 0.071B + 128 \end{aligned} \tag{4}$$

Hue and saturation are calculated through Equations (1)–(3), and Y, Cb, and Cr are calculated through Equation (4). Y means luminance, Cb means blue-difference, and Cr means red-difference.

According to Rumelhart et al. [27], the process of establishing the connection strength of a neural network was called learning. The error signal ($\delta$) was calculated using the difference between the output calculated by the input and the expected output of the input, and the process of correcting the connection strength by propagating the error signal to the previous layer was repeated. At this time, the process of propagating the error signal to the previous layer was called the backpropagation algorithm, and neural network learning was performed through this process. In the backpropagation

algorithm, the connection strength is changed by the error signal, and if the learning rate is set small, the amount of connection strength variation is reduced, and the learning process is slowed down. In order to prevent the learning process from being slowed down, the variation of the previous learning phase is supplementary used when the connection strength is changed. This method is referred to as a momentum backpropagation algorithm [28].

$$v^{new} = \frac{\beta \, v^{old}}{\|v^{old}\|}, \ \beta = 0.7^n \sqrt{p}, \tag{5}$$

For initial connection strength setting, the initial connection strengths were set in the range of $w = -0.5$ to $+0.5$ and $v = -0.5$ to $+0.5$ according to Nguyen and Widrow [29]. Connecting strength $v$ was initialized by Equation (5) at the start of learning. In Equation (5), $n$ is the number of neurons in the input layer and $p$ is the number of neurons in the hidden layer. The method of Mirchandani and Cao [30] was used to determine the number of hidden neurons. The maximum number M of linearly separable regions was calculated as Equation (6); $n$ is the number of dimensions of the input pattern space and $p$ is the number of neurons in the hidden layer. In Equation (6), *if k is less than $p$ and $n$ is larger than or equal to $p$*, then M was calculated as Equation (7).

$$\text{M} = \sum_{k=0}^{n} {}_pC_k, \tag{6}$$

$$\text{M} = {}_pC_0 + {}_pC_1 + \cdots + {}_pC_p = 2^p, \tag{7}$$

the number of neurons in the hidden layer to solve the problem of the linearly separable region, p, was calculated by Equation (8).

$$p = \log_2 M, \tag{8}$$

$$\begin{aligned} \Delta v^k &= \alpha \, \delta_z x^k + \beta \Delta v^{k-1} \\ \Delta w^k &= \alpha \, \delta_y z^k + \beta \Delta w^{k-1} \end{aligned} \tag{9}$$

The learning rate a was changed within the range of 0.001–0.1, and it was set with reference to the progress of learning. In this study, a unipolar sigmoid function was used. This is because the size of the image information transmitted as the input was normalized to a size of 0 to 255. In Equation (9), $\alpha$ means learning rate and $\beta$ means momentum constant. $\delta_z$ denotes the error signal of the hidden layer, and $\delta_y$ denotes the error signal of the output layer. In this study, the momentum constant was set to 0.8 [28].

The expected output value was set to [0 0 1] for red sweet pepper, [0 1 0] for yellow sweet pepper, and to [1 0 0] otherwise. A total of 204,800 pixels were used for the extracted color information and 184,320 pixels were used for the learning. The remaining 10% were used to confirm the learning outcome and the learning was terminated if the learning outcome was over 99%. In order to check the learning result, this study used the learned weights $v$ and $w$ to output the image information, which was transmitted through the input layer into the output layer as images. When the output layer activation state was [0 0 1], red pixels were displayed in the image, yellow pixels were displayed in [0 1 0], and green pixels were displayed in the other cases. The activation state of the paprika region was confirmed by comparing the output image of the activated paprika and the input image of the paprika cultivation environment.

*2.3. Vison Servo Control*

Since an image processing-based harvest uses a non-contact measurement method, it is appropriate for robot harvest works. The visual detection and control can be seen through the open-loop mode which "sees" and "moves." In addition, the accuracy of works depends on the accuracy of the image sensor and the accuracy of the end-effector control [20]. As an alternative to enhance the accuracy

of the servo system, this study used a visual-feedback control loop, and increased the accuracy of the system. The mechanical vision can provide a closed-loop position for the robot's control of the end-effector and this mechanism is referred as a vison servo system.

As shown in Figure 9, an eye-in-hand vision servo system was used in this study. The eye-in-hand vision servo system is a type of camera that is fixed at a robot's end-effector. In this method, there are already known and fixed relations between the poses of the camera and end-effector. In this case, the pose relation between the end-effector and a camera is referred to as $X_c^e$, and the pose relation between an object and a camera is referred to as $X_t^c$. The image-based control method was used to control the pose of the end-effector approaching the object. According to Corke [31], in the image-based vision servo, control is carried out by the position and pose of an object that exists in the image.

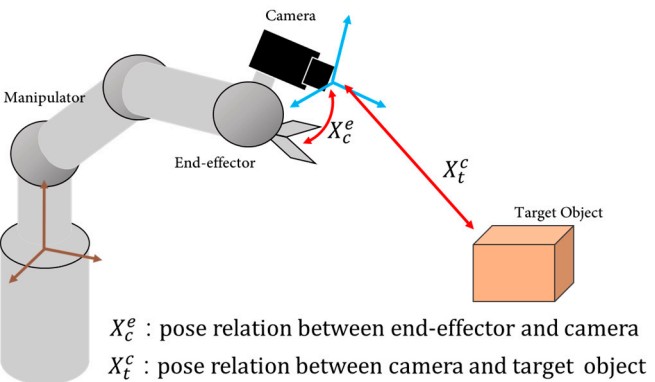

**Figure 9.** Eye-in-hand vison servo system.

In order to realize a vision servo-type sweet pepper harvest algorithm, this study acquired the center of a sweet pepper through the camera installed at end-effector, and then controlled the end-effector to be matched with the said sweet pepper, thus extracting feature points and adjusting stereo matching, and then measuring distances. After acquiring a three-dimensional coordinate, the end-effector approaches a sweet pepper based on the information acquired and then detects its position through an image, thus manipulating the end-effector.

Vision servo system's working processes for the automated harvest of sweet pepper are shown in Figure 10. The center of the sweet pepper was first extracted through Cam3 installed in the end-effector, and the end-effector was controlled so that the direction of the end-effector coincided with the direction of the sweet pepper. Then, the distance was measured through stereo matching. Based on the 3D coordinate information measured by Cam1 and Cam2, after the end-effector approaches the sweet pepper, the posture of the end-effector is controlled while confirming the position of the fruit by the image of Cam3. After that, the vision servo system checks the image of Cam3 and repeats the process so that the peduncle part of the sweet pepper approaches the cut part of the end-effector.

When the ROI (region of interest) of the sweet pepper is extracted, the manipulator rotates to place the center of the ROI in the center of Cam3, thus controlling the Y-axis manipulator and then making the center of Cam3 matched with the bottom of the ROI. The reason for this process is to allow Cam1 and Cam2 to acquire images of sweet pepper fruit during stereo matching. Experiments were conducted to confirm that the center of the camera and the center of the paprika were controlled to coincide with each other and confirmed the results. The results were analyzed by measuring the unit of pulse for controlling the manipulator, the rotation angle of the rotation axis manipulator, and the movement distance of the vertical axis manipulator.

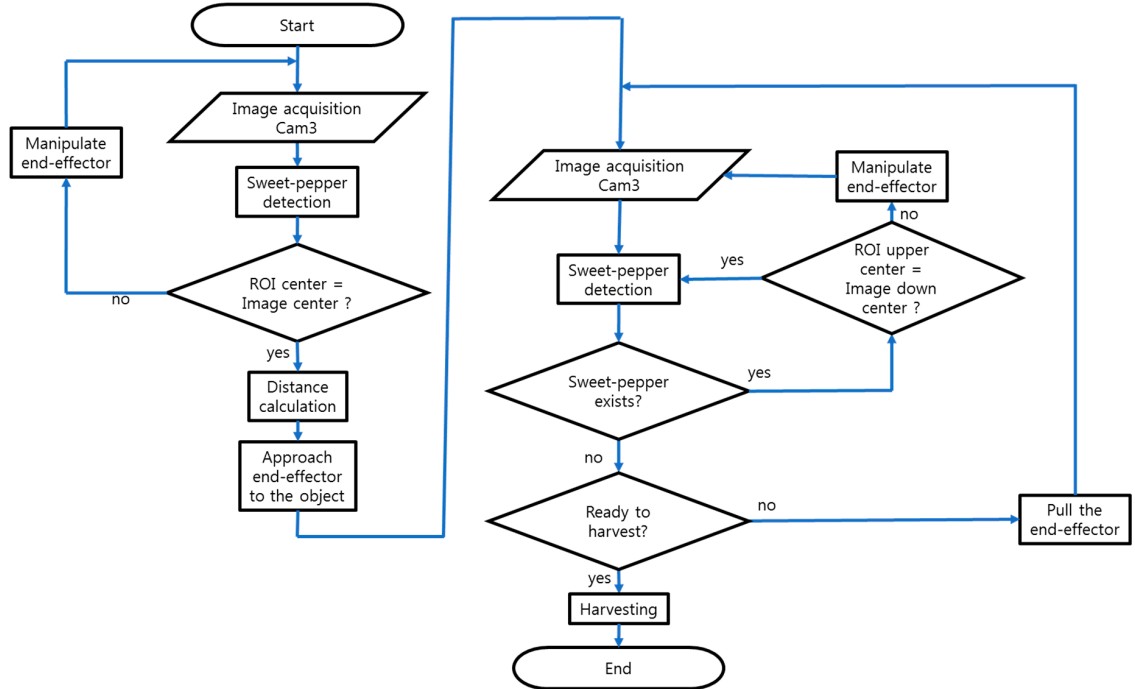

**Figure 10.** Flow chart of harvesting algorithm.

When the center of the ROI and the center of Cam3 are matched, the system extracts the source of laser light irradiated and then carries out the adjustment of irradiated points, thus measuring distances. It extracts green laser that exists in the sweet pepper and then examines the original form of the laser. In case the form is circle, it extracts the center of the circle and carries out adjustments on the basis of the center of the circle. When it carries out the adjustment, it uses epipolar geometry [32]. After examining the disparity of adjusted points, it calculates the z-score and then classifies the outlier. Using disparity examined and calibration results, it measures distances and then moves the horizontal manipulator, thus placing the end-effector in the bottom of the sweet pepper (Figure 11).

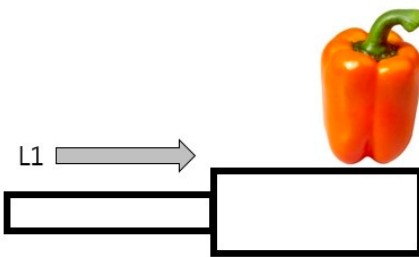

**Figure 11.** Approaching sequence.

As shown in Figure 12, the peduncle of the sweet pepper is fixed in stems or branches. When it moves horizontally, therefore, it rotates on the axis of the peduncle. If the end-effector moves vertically as much as the sweet pepper rotates in this process, the peduncle can be positioned in its cutting portion. At this time, it is necessary to use the axis of rotation and then make the peduncle continuously placed in the center of the end-effector. As shown in Figure 13, this study created a similar environment to carry out the experiment of establishing control variables for the control of vision servo using the model of sweet pepper.

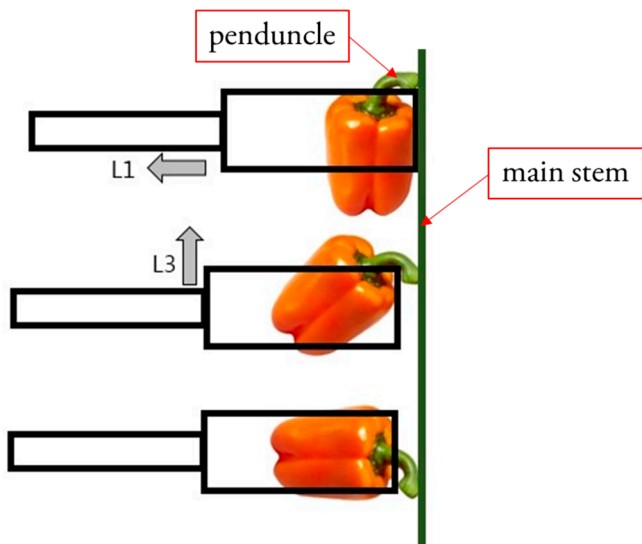

**Figure 12.** Sweet pepper pose control.

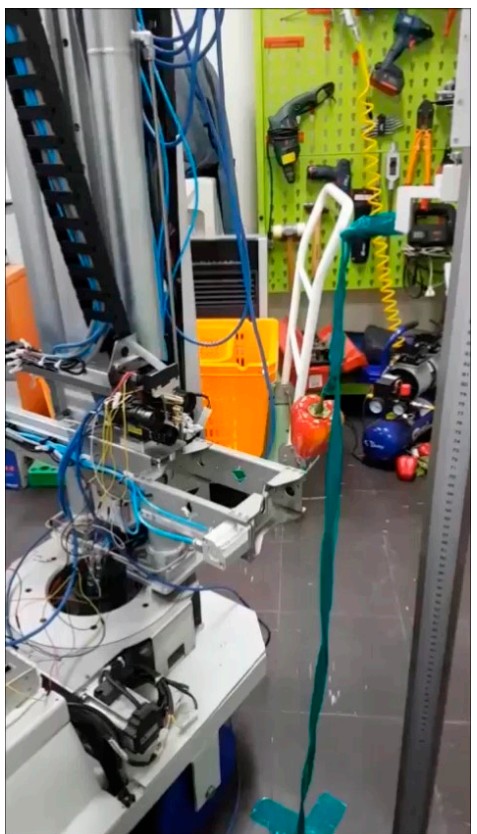

**Figure 13.** Set-up of control variables.

In order to realize a vision servo-used harvest algorithm, the control variables of the manipulator were established according to the position and pose of objects appearing in stepwise harvesting images. Figure 14 presents the ROI of a sweet pepper detected from the image, and the difference between the position of the center of the bottom and the center of the camera. After analyzing the difference of X and Y directions in the image, the rotation control variable θ2 was established based on the difference of the X direction, and the vertical control variable L3 was established based on the difference of the Y direction.

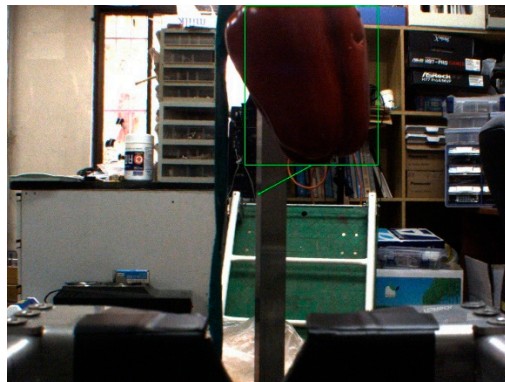

**Figure 14.** Detected sweet pepper's region of interest (ROI) and camera center.

*2.4. Harvest Experiment*

As a field experiment, this study conducted a harvest experiment with sweet peppers planted in a greenhouse of the Gumi Horticulture Confederation, Gyeongbuk Province in South Korea. Figure 15 is a scene of the field experiment. The field experiment was carried out to measure the errors of control variables (rotation axis error (°), vertical axis (mm)) in the process of detecting a sweet pepper and of making it match with the center of the end-effector. The errors by distance measurements were also measured. The success of harvesting by peduncle thickness and length was analyzed.

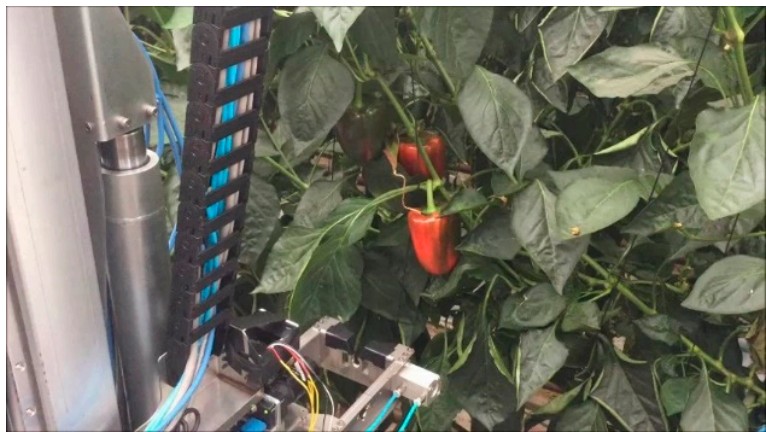

**Figure 15.** A scene of the field experiment.

The harvesting success rate for the thickness of the peduncle was analyzed in three cases (0–10 mm, 10–15 mm, and more than 15 mm). The harvesting success rate for the length of the peduncle was analyzed in three cases (0~3 mm, 3~6 mm, 6~9 mm, and more than 9 mm). Sessions of experiment were repeated four times, thus checking the efficiency of the harvest.

## 3. Results and Discussion

*3.1. Sweet Pepper Recognition*

Momentum backpropagation algorithm was performed by inputting H, S, Y, Cb, Cr, R, G, and B of the fruit of red and yellow sweet peppers and other areas. Weights were calculated by learning 1 million hidden neurons and 3 hidden neurons with a maximum error of 0.05, learning rate of 0.01, and momentum constant of 0.8. Figures 16 and 17 show the result of extracting the fruit region of sweet peppers through the momentum backpropagation algorithm. Figures 16 and 17 show the output values calculated using the learned $v$ and $w$ values as red, yellow, and green pixels. A red pixel means a pixel recognized as a fruit region of red pepper, a yellow pixel means a fruit region of a yellow

pepper, and a green means a pixel recognized as another region. It was confirmed that a lot of noise is generated in areas where it is difficult to extract the intrinsic colors such as reflections and shadows generated by illumination, but the actual paprika regions are clearly extracted. Figure 18 shows the fruit area of the pepper in green squares in Figures 16 and 17.

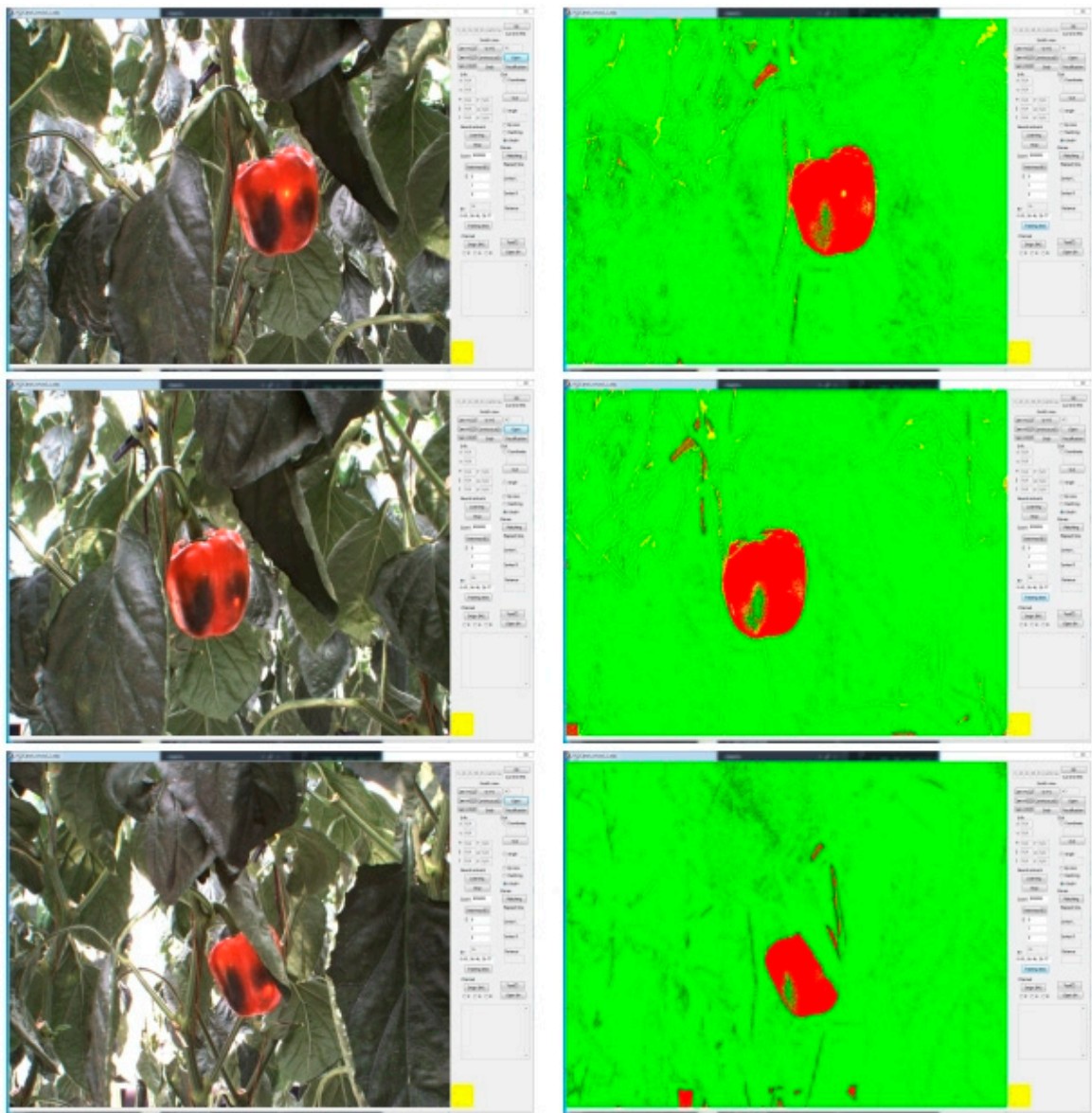

**Figure 16.** Activation images using the backpropagation algorithm (red sweet pepper).

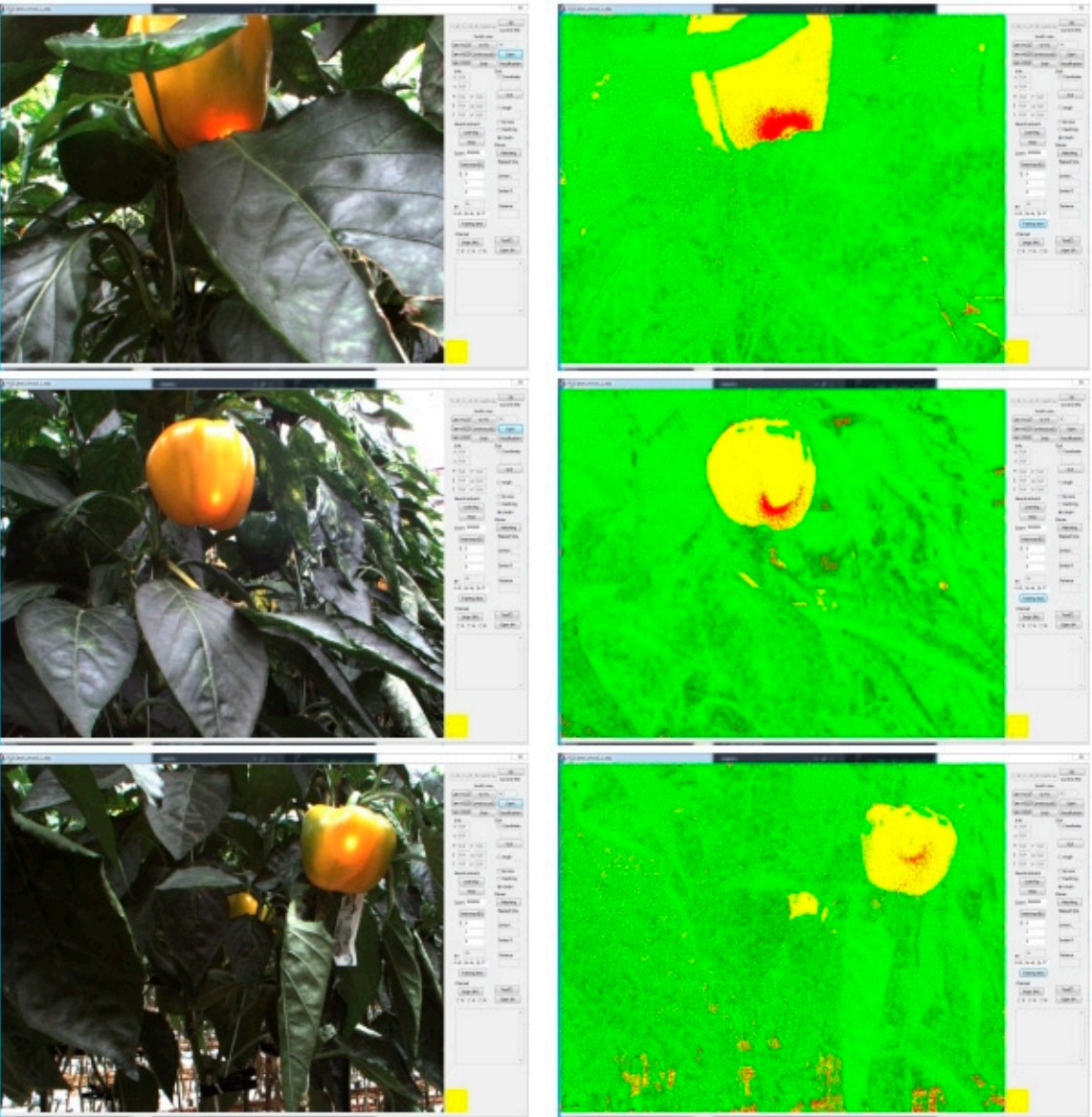

**Figure 17.** Activation images using the backpropagation algorithm (yellow sweet pepper).

In order to confirm the performance of the sweet pepper recognition in this study, 269 sweet pepper images were used to extract fruits. Out of 269 cases, 221 (82.16%) sweet pepper images were recognized successfully. As a result, 26 cases (9.67%) were recognized as one fruit when two or more paprika were attached. When the difference of contrast was severe, the recognition failure occurred once (0.37%). Among the cases of recognition failure, 21 cases (7.81%) were when there were many areas obscured by leaves and branches.

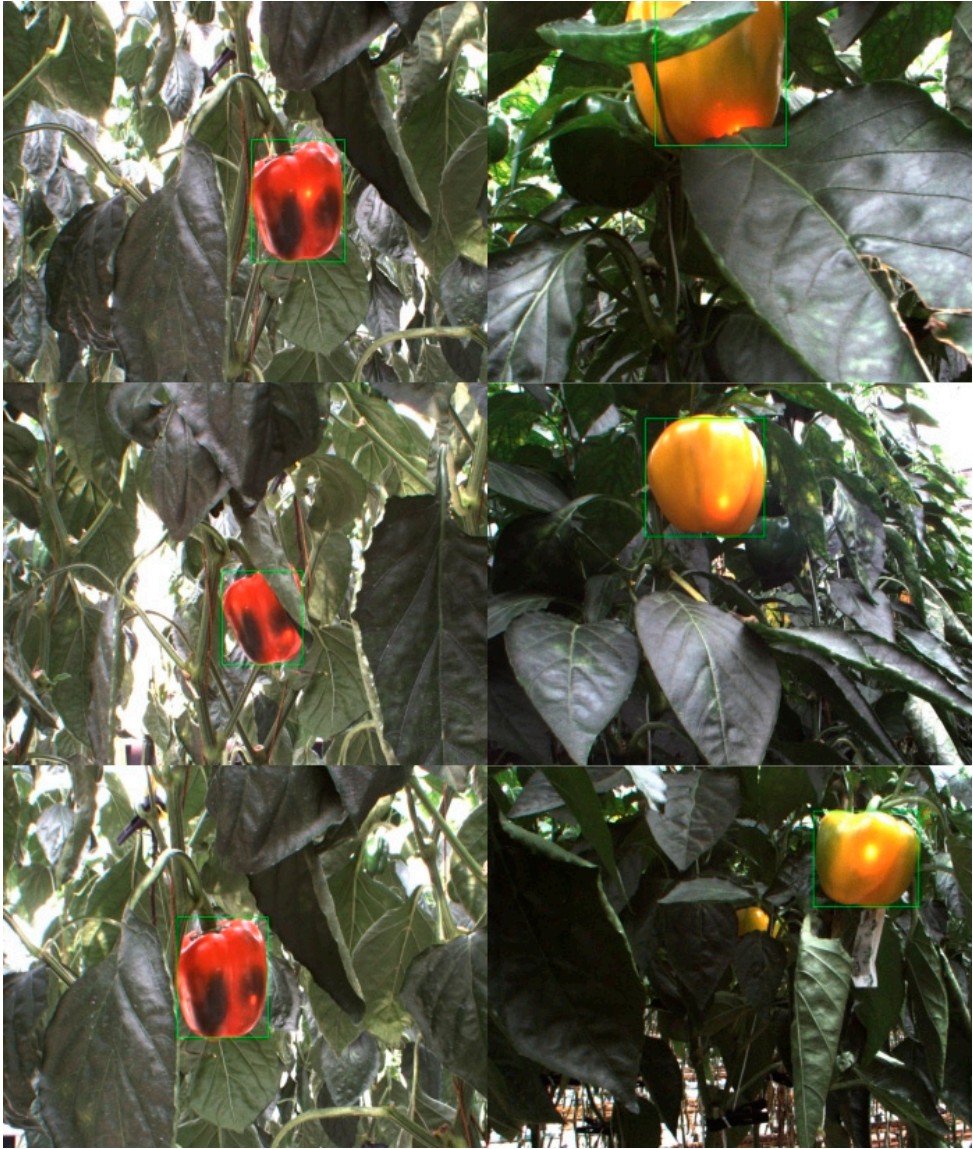

**Figure 18.** Extracted sweet peppers.

### 3.2. Set-Up of Control Variables

An experiment was carried out to make the center of the sweet pepper detected from the image of the sweet pepper that existed in 350~600 mm distance matched with the center of camera, and then use the necessary control variables to control the manipulator to directions of vertical (Y) and horizontal (X). The rotation axis was made to rotate by 2.4° per 10,000 pulses, and the horizontal axis was made to be controlled by 10 mm per 10,000 pulses.

The size of a pixel corresponding to a length of 30 mm at a distance of 350 mm and a distance of 600 mm was measured. The position of the changed pixel was calculated according to the angle change. When the distance from the camera was 350 mm, the number of pixels occupied by the 30 mm length line in the image was 342 pixels. When the distance from the camera was 600 mm, the number of pixels occupied by the 30 mm length line in the image was 127 pixels. The actual length of one pixel at each distance was 0.088 mm at a distance of 350 mm and 0.236 mm at a distance of 600 mm. When 1 pixel changes in 350 mm distance, $\theta2$ must be controlled by 0.014°. On the other hand, when 1 pixel changes in 600 mm distance, $\theta2$ must be controlled by 0.023°. However, it was not possible to control the information of distance in the experiment, and therefore the angle was controlled only

by the average 0.018° for 1 pixel. Since the rotating axis can rotate 2.4° per 10,000 pulses, 75 pulses must be given to rotate 0.018°. In other words, 75 pulses for 1 X-axis pixel are used to control θ2. As it approaches the center of the picture and the difference of X-axis pixels is less than 15, 50 pulses are used to control the pose accurately. As for the Y-axis control based on correlation between pixels and distances presented in Table 1, change of 1 pixel in an image is 0.088 mm on 350 mm distance, and 0.236 mm on 600 mm distance. As in the control of the X-axis, an average of 0.16 mm per pixel are used to control L3. A total of 160 pulses per Y-axis pixel was used to control L3.

**Table 1.** Variation of coordinate.

| No. | 90 mm | 110 mm | 130 mm | 150 mm |
|---|---|---|---|---|
| 1 | 127 | 109 | 101 | 102 |
| 2 | 126 | 108 | 105 | 98 |
| 3 | 143 | 109 | 104 | 102 |
| 4 | 128 | 107 | 101 | 100 |
| 5 | 134 | 104 | 106 | 100 |
| 6 | 108 | 106 | 102 | 98 |
| 7 | 123 | 105 | 101 | 97 |
| 8 | 125 | 104 | 98 | 94 |
| 9 | 130 | 107 | 103 | 99 |
| 10 | 128 | 105 | 102 | 101 |
| Average | 127.2 | 106.4 | 102.3 | 99.1 |

As a result of the control experiment using the set control variables, it was confirmed that the error diverges without converging to 0 based on ±30 pixels. Therefore, when the Y-axis pixel errors were less than 30, 100 pulses per pixel were used to control.

Since harvest was carried out with the sweet pepper that exists within the distance of 90~150 mm from the camera installed at the end-effector, the change of distance per pixel in that scope of distance was measured.

Table 1 presents the change of the image coordinate for the 10 mm change of spatial coordinate from the distances of 90, 110, 130, and 150 mm, through the camera of the end-effector. The table presents that the difference of measurement from the distance of 90 mm is bigger than that from any other distances. This difference seems to be caused by the distortion of the image that is taken at a close distance. Therefore, on the basis of 10 mm/100 pixels, L3 for the control of the Y-axis in three steps were established at 100 pulses/per pixel. In addition, in order to convert the result to angles, it is necessary to have the value of the manipulator L1 variable.

$$\alpha = 2\tan^{-1}\left(\frac{1}{L1 + 230}\right), \tag{10}$$

Equation (10) presents the calculation of angle $\alpha$ for the change of 1 mm X coordinate using L1 variable. Angle $\alpha$ is used to calculate the control pulse of θ2 according to the angle per pulse.

Table 2 presents the result of the experiment. Repeated 20 times, the table presents the errors of the X-axis and Y-axis, together with the times of control executed in every experiment. In order to detect the position of the sweet pepper from the image and then move the end-effector to the target position, it was necessary to carry out an average 4.15 times of controlling process.

**Table 2.** Test results of a sweet pepper's position controlling.

| No. | Error x (pixel) | Error y (pixel) | Control Count |
|---|---|---|---|
| 1 | 0 | 5 | 3 |
| 2 | −1 | 7 | 4 |
| 3 | −1 | 0 | 4 |
| 4 | −6 | 10 | 3 |
| 5 | −10 | 5 | 3 |
| 6 | 9 | 4 | 3 |
| 7 | −2 | 7 | 3 |
| 8 | 1 | −3 | 8 |
| 9 | 3 | 1 | 3 |
| 10 | 6 | 3 | 5 |
| 11 | −2 | −5 | 3 |
| 12 | −5 | 5 | 3 |
| 13 | −5 | 2 | 4 |
| 14 | 3 | −9 | 6 |
| 15 | −3 | 9 | 6 |
| 16 | 4 | 2 | 7 |
| 17 | −9 | 1 | 5 |
| 18 | −2 | 7 | 3 |
| 19 | 2 | 3 | 3 |
| 20 | 9 | 1 | 4 |
| Average | −0.45 | 2.75 | 4.15 |

Figure 19 presents the change of errors after undergoing the controlling process eight times. It confirms that from the fifth control, errors approach to 10 pixels. This result seems to derive from the fact that convergence is not realized by the error occurred in the extraction of the ROI of the sweet pepper. Average control errors were −0.45 pixels for the X-axis and 2.75 pixels for the Y-axis. Additionally, 2.75 pixels can be converted to 0.65 mm in 600 mm and therefore there seems to be no problem in the control of the vision servo.

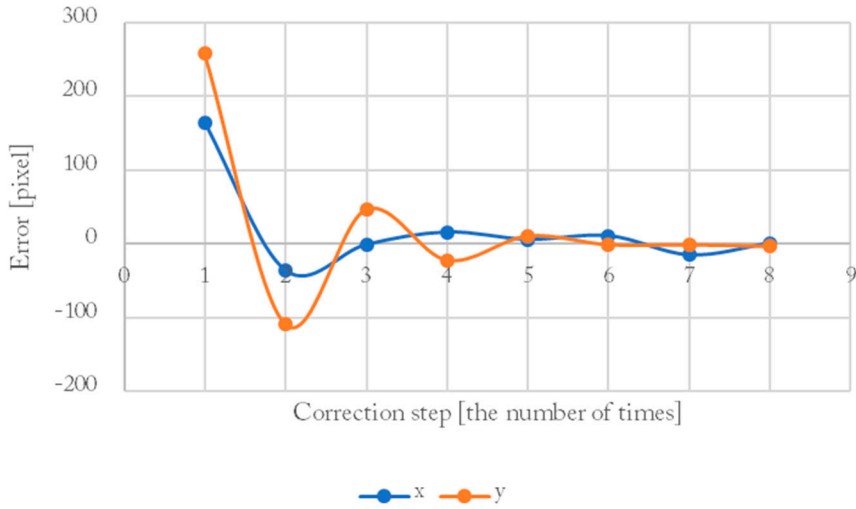

**Figure 19.** The error of pixels in the image according to the number of repetitive controls that occurs when moving the end-effector to the target point.

### 3.3. The Result of Harvest Experiment

The harvest experiment was carried out in three steps. After completing each step, results were examined.

Figure 20 presents the result of realizing the algorithm to move the position of the end-effector to the center of the harvesting sweet pepper using the rotation axis, and of practicing the algorithm. In order to calculate the real coordinates, the sweet pepper that appeared in the image was checked and then the pulse of the manipulator was inputted by 500 units, thus matching centers and calculating angles using distances. After moving the coordinate of the end-effector based on the algorithm, the pulse measured was presented. The error of angle was 1.2° in maximum and 0.77° on average, and standard deviation of errors of angle was 0.2652. The distance-based error was 11.3 mm in maximum and 6.22 mm on average, and standard deviation of distance-based errors was 2.0368. When the error of the X-axis was 11.3 mm, the horizontal length of the end-effector was 120 mm and the average diameter of the sweet pepper was 80.92 mm.

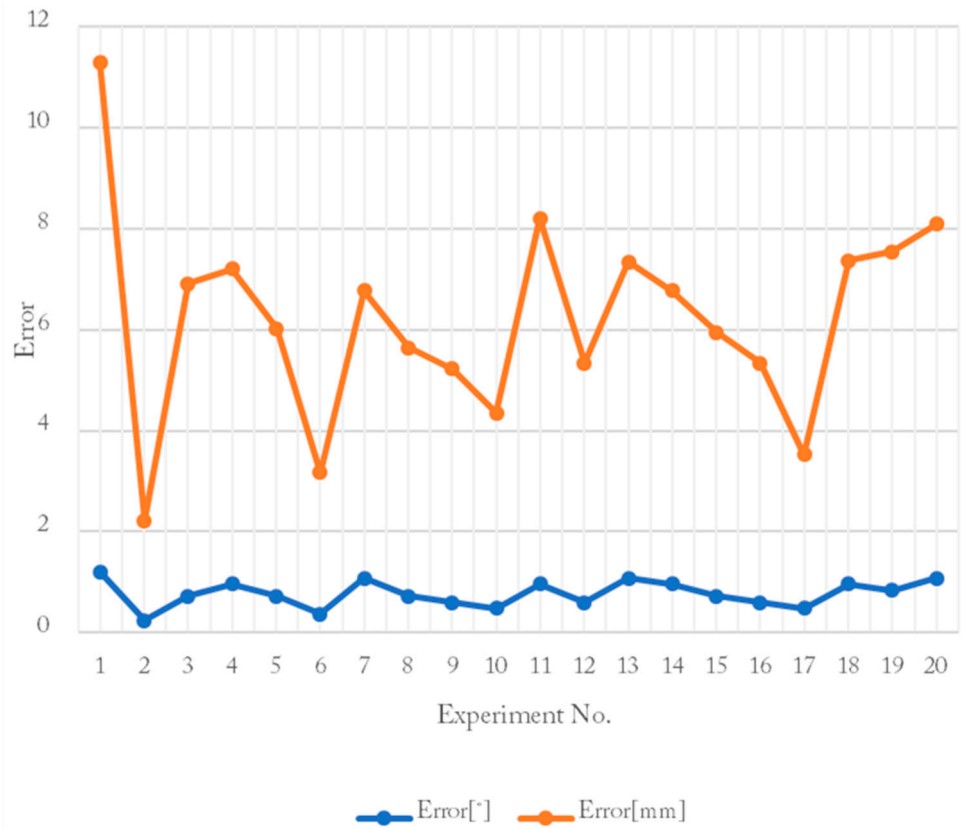

**Figure 20.** Field experiment results of sweet pepper's position extraction algorithm (rotation axis).

Figure 21 presents the result of realizing the algorithm to move the position of the end-effector to the bottom of the harvesting sweet pepper using the vertical axis manipulator, and of practicing the algorithm. The maximal error was 17.1 mm and the average error was 8.205 mm, and standard deviation was 5.5497. The reason why the maximal error was bigger than the horizontal error seems to be derived from the fact that the aspect ratio of the image was small and therefore the number of pixels was relatively small as well. In the process of searching for the position of the sweet pepper and moving the end-effector to the target position, there occurred a maximum 11.3 mm error horizontally, and maximum 17.1 mm error vertically. The error occurred in the detection of sweet pepper and the set-up of the ROI generates the control error, and position was revised by extracting the pose of sweet pepper and by carrying out harvesting works.

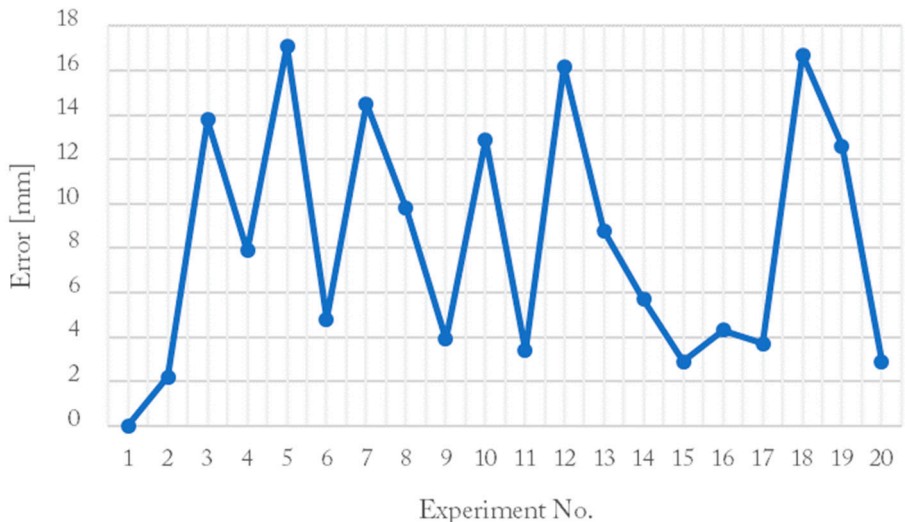

**Figure 21.** Field experiment results of sweet pepper's position extraction algorithm (vertical axis).

The results of the experiment to acquire information on the depth from the system to the sweet pepper, namely the second step harvest algorithm, is presented in Figure 22. Z means a real coordinate of distance and signifies a calculated coordinate of distance. The result of executing the algorithm of extracting information on depth presents the fact that the maximal error was 39 mm and the average error was 10.05 mm, and standard deviation was 12.1595. Compared with the result of indoor experiments of which maximal error was 31 mm and average error was 8.44 mm, the errors were bigger. These bigger errors seem to be because the natural light interferes with the laser light source that was used to extract feature points.

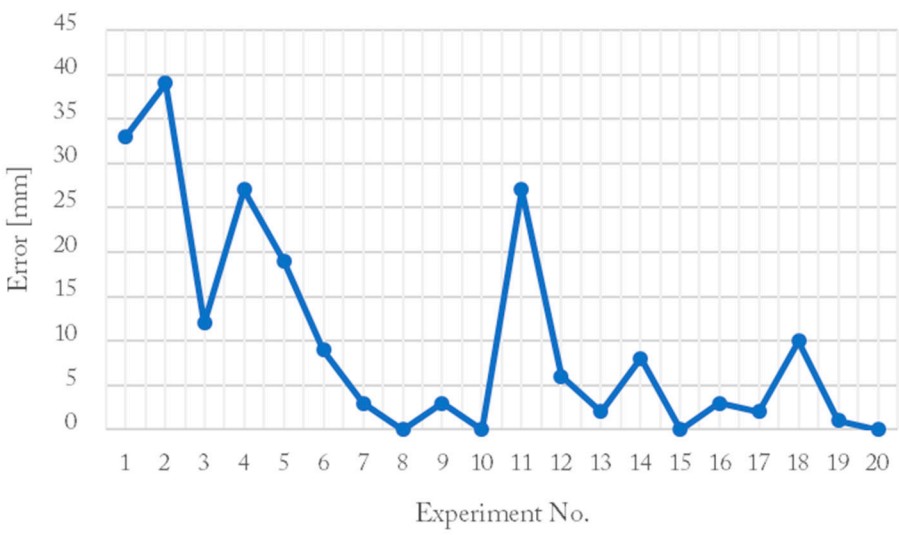

**Figure 22.** Field experiment results of distance compute algorithm.

The process of transferring the end-effector to harvest sweet peppers was examined by classifying it into the thickness, length, and angle of the peduncle, in order to make the end-effector approach to the portion of cutting.

The thickness and length of the peduncle were calculated on the basis of the end-effect's contact point of cutting, and the vertical angle of the peduncle to the system was set up at 0°. The thickness of the peduncle was classified by its diameter of 0~10 mm, 10~15 mm, and more than 15 mm, thus examining their rate of successful harvest. As for the diameter, the portion that has contact with the

end-effector's cutting point was measured. The most frequent diameter of the sweet pepper's peduncle appeared in the scope of 10~15 mm. The number of experiments was 30 times, and 16 of them were successfully harvested. The scope of 10~15 mm was most successful with 60.0% of 9 successes in 15 trials. The success rate of the scope of 0~10 mm was 58.3% of 7 successes in 12 trials. In the scope of 15 mm and above, it was impossible to harvest. The failure of harvest in the scope seems to be because the diameter of peduncle was bigger than the entrance of the cutting portion.

The rate of successful harvest according to the length of the peduncle is presented in Table 3. The rate was examined by dividing the length of the peduncle into 0~3 mm, 3~6 mm, 6~9 mm, and more than 9 mm. Most of the harvest of sweet peppers where the length of the peduncle was less than 6 mm was unsuccessful. It was seemingly because the guide installed in front of the end-effector was too thick.

**Table 3.** Field experiment results of harvesting algorithm according to length of peduncle.

| 0–3 mm | | | 3–6 mm | | |
|---|---|---|---|---|---|
| **Try** | **Success** | **Rate** | **Try** | **Success** | **Rate** |
| 2 | 0 | 0% | 5 | 1 | 20% |
| **6–9 mm** | | | **~9 mm** | | |
| **Try** | **Success** | **Rate** | **Try** | **Success** | **Rate** |
| 15 | 9 | 60% | 8 | 6 | 75% |

The success rate of harvesting according to diameter and length of peduncle was analyzed and the overall success rate was 53.3%. The approximation rate of the end-effector to the peduncles of sweet pepper was 86.7%. When the same harvesting procedure was repeated four times using the sweet pepper which failed to harvest, the maximum harvesting success rate of about 60 sweet peppers was 70%.

## 4. Conclusions

In this study, the shape of the manipulator for the automated robot system to harvest sweet peppers was cylindrical (consisting of a rotation axis, a horizontal axis, and a vertical axis), because it was efficient to carry out harvesting works in a narrow space. Based on the external dimension of a sweet pepper, the end-effector was designed not to damage the stem but to approach it from below, thus harvesting it without damage. It was also designed not to check the peduncle additionally but to use information on the fruit only, and its cutting portion was designed to be vertical, thus preventing the cutting of the stem. The study used three color CCD cameras, consisting of a stereo vision system and an eye-in-hand image system for the control of the end-effector's vision servo. In addition, using laser light sources, it was equipped with a light system that could add feature points to the sweet pepper. Using a camera installed at the end-effector, it could also detect a sweet pepper, and using the upper and bottom center of the ROI, it could extract the position and pose of the sweet pepper. Based on the information extracted, the three-step harvest algorithm was developed. In order to confirm the performance of the sweet pepper recognition in this study, 269 sweet pepper images were used to extract fruits. Of 269 sweet pepper images, 82.16% were recognized successfully. The result of the experiment with 60 sweet peppers presented the fact that its approach rate to peduncle was about 86.7%, and via four sessions of repetitive harvest experiment it achieved a maximal 70% harvest rate, and its average time of harvest was 51.1 s. According to a study by Joung et al. [33], the average number of harvest days for red sweet pepper 'cupra' in South Korea was 81.9 days and the fruit yield per 10 are was 18,848 kg. In order to do that, when one farmer works eight hours a day, 143 red sweet peppers (200 g per fruit) should be harvested in one hour. In order to replace the manpower by using this study, where the harvesting success rate is 70% and the one-time working time is 51.1 s, and by assuming that the harvesting success of 70% on 205 paprika is obtained and assuming the

system works eight hours a day like manpower, the three systems must perform. However, since the system does not require rest, unlike humans, it is assumed that if the system works 24 hours a day, it will get the same daily yield as the manpower. Compared with the study by Barth et al. [19] in the Netherlands, their sweet pepper harvesting robot showed 85% sweet pepper recognition rate and 70% harvest success rate. The recognition rate was 2.84% lower than that of Barth et al. who detected fruits of sweet pepper using the convolution neural network. There were no differences in the harvest success rate though environmental differences [19,22] such as spacing distance of plant row and crop density in South Korea and the Netherlands. If the volume of the front portion of the end-effector of this study decreased, and if the design of the cutting portion of this study improved, and if the pneumatic cylinder of this study was placed at the back of the end-effector, then the rate of harvest would increase. In addition, if the motor RPM (Revolutions per minute) of the manipulator of this study increased, the time of harvesting would seemingly decrease.

**Author Contributions:** All five authors contributed almost equally to all aspects of this research. Conceptualization, B.L., D.K., B.M., J.H. and S.O.; methodology, B.L. and J.H.; software, B.L. and J.H.; validation, B.L., D.K., B.M., J.H. and S.O.; formal analysis, B.L. and J.H.; investigation, B.L., D.K. and J.H.; resources, B.L., D.K., B.M., J.H. and S.O.; data curation, B.M. and S.O.; writing—original draft preparation, B.L., D.K. and J.H.; writing—review and editing, B.L., D.K. and J.H.; visualization, B.M. and S.O.; supervision, B.M. and S.O.; project administration, B.M. and S.O.; funding acquisition, B.M. and S.O.

**Funding:** This research was funded by the Korea Institute of Planning and Evaluation for Technology in Food, Agriculture, Forestry and Fisheries, Ministry of Agriculture, Food and Rural Affairs (MAFRA), grant number 114046-3.

**Conflicts of Interest:** The authors declare no conflict of interest.

## Nomenclature

| | |
|---|---|
| $X_c^e$ | The pose relation between end-effector and a camera |
| $X_t^c$ | The pose relation between a camera and an object |
| $\theta2$ | Angle for the change of X |
| $\alpha$ | Calculated angle of control pulse |

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
