# Peer review of "A Vision Servo System for Automated Harvest of Sweet Pepper in Korean Greenhouse Environment"

_applsci, doi:10.3390/app9122395_

Round 1

Reviewer 1 Report

Note 1

Term: Visual Servo control (image-based closed-loop control)

It seems to be inadequate to the described solution.

The word “visual” means - visually - made with the use of sight (eyes).

The word “vision” means made using a camera or a vision sensor, e.g. a machine vision system, a robot vision system

I suggest:  “Vision servo control”

Note 2
Figure 1.   Figure 1 has a and b section which is not explained in the drawing's signature. In addition, in both drawings the same rotation angle has been described once as ϴ1 and then as ϴ2.

Note 3 
Line 101
Visual Servo System. change to  “Vision servo control”

Line 188. Visual Servo System. change to  “Vision servo control”

Note 4

Drawings such as figure 6 should be replaced with an engineering drawing. Please set the coordinate systems and relations between the systems. Please define in the technical drawing all set parameters of the robot's work. (Fig. 4), (Fig. 6)

Note 5

Figure 8 shows the fragment concerning the analysis of the image from the camera No. 3. However, the engineering drawing with camera positions and their numbering has not been defined before

The article lacks a work diagram and data exchange between system elements.The data of the cameras used to build the robot is missing, there is no data about the data transmission and data conversion protocol.

There is no data on the time of robot position control when positioned on the basis of camera data.

Note 6

The work lacks a professional description of the on processing algorithm. The lack of a detailed description makes it impossible to assess the concept adopted by the authors to complete the task. Please provide a detailed description of the image processing algorithm together with the data flow diagram in the robot control system.

Interesting subject matter, however, with very poor technical description. There are no technical drawings, wiring diagrams in the control system and data exchange between devices.

There is also a lack of description of the image processing algorithm taken step by step. This makes it impossible to evaluate the solution.

In the experimental data, no image recorded by the cameras at the time of the harvest was included. Only the pepper on the background of the laboratory is visible,

Author Response

We have responded to editor and reviewers’ comments and revised our manuscript based on their recommendations. We believe that these corrections can give clear understanding to the reviewers and readers and improve the quality of our manuscript. Please, find the attached files containing the revised manuscript and the responses to the reviewers’ comments.

We look forward to seeing a publication in the Journal of the Applied Sciences pretty soon.
Thank you for your kindness.

Kind regards,
Se-Bu Oh

Reviewer 2 Report

Dear Authors,

I revised the manuscript “A Visual Servo System for Automated Harvest of Sweet-pepper” submitted to the Applied Sciences Journal. The paper is very interesting. However, I have some concerns, which need to be addressed before considering for final publication.

Line 35. Use [2,3].

Line 45. Use [4-7].

Line 50. I suggest adding chapter “2. Material and methods” and assigning a new number to subsection “2.1 Hardware Composition”.

Line 55. Fig. 1. The height “U” is missing in the figure 1.

Line 63. Fig. 2. Poor figure quality.

Line 70,72. What does this "kgf" unit mean? Describe in more details.

Line 71. The values given as mm/s please change to mm∙s-1 .

Line 93-94. Format the equation as required in the file “applsci-template.dot” in section 3.3. Formatting of Mathematical Components.

Line 101. I suggest assigning the number 2.2 (if you accept my suggestion from line 50).

Line 160. I suggest assigning the number 2.3 (if you accept my suggestion from line 50).

Line 174. I suggest assigning the number 3 (if you accept my suggestion from line 50).

Line 175. I suggest assigning the number 3.1 (if you accept my suggestion from line 50).

Line 195-196. Format the equation as required in the file “applsci-template.dot” in section 3.3. Formatting of Mathematical Components.

Line 201. Format the equation as required in the file “applsci-template.dot” in section 3.3. Formatting of Mathematical Components.

Line 218. Format the equation as required in the file “applsci-template.dot” in section 3.3. Formatting of Mathematical Components.

Line 238. Use square brackets “[]” for units.

Line 249. Use square brackets “[]” for units.

Line 260. Use square brackets “[]” for units.

Line 287. I suggest assigning the number 4 (if you accept my suggestion from line 50).

Line 315. If you don’t attach “Supplementary Materials”, delete this line.

Line 316. “Author Contributions” is missing.

Line 319. Add more references (about 5-10), especially for the section "Introduction" and "Conslusion".

Author Response

(The authors gave the same response as above.)

Reviewer 3 Report

The authors describe visual servo system for harvesting sweet peppers with a robot. They conducted field experiments in a greenhouse to evaluate their visual servo control system. Unfortunately, the paper is not well written, and does include several errors which makes it not suitable for publication.

The Introduction is not suited to the work described in the paper, the state of the art is not described.

The Materials and Methods part is bad. It is completely unclear how the robot detect sweet peppers, how the algorithm work and why you need 3 cameras instead of one.

The Images are of bad quality and do not help to understand the work.

The evaluation method is not well described.

Therefore, I suggest to reject the paper.

Some more comments:

Line: 40 why does the environment must be complex? Maybe an ordered sweet-pepper greenhouse can be a simple environment?

Introduction is too short. No mentioning of the state of the art

Figure 2: fuzzy, please replace

Figure 4: what does the circles mean?

Line 87: where does this average diameter comes from?

Formula (1) what is the meaning of the numbers?

Line 102: why bold letters?

Figure 7 is not necessary, describe in text

Figure 8:  lines are not straight, hard to follow the flow, decision boxes are not clear

Author Response

(The authors gave the same response as above.)

Round 2

Reviewer 1 Report

The authors made significant corrections and included reviewers' suggestions.

Author Response

We have responded to editor and reviewers’ comments and revised our manuscript based on their recommendations. We believe that these corrections can give clear understanding to the reviewers and readers and improve the quality of our manuscript.

We appreciate your kindness comment.

We look forward to seeing a publication in the Journal of the Applied Sciences pretty soon.

Thank you for your kindness.

Kind regards,

Se-Bu Oh

Reviewer 3 Report

The quality of the paper increased a lot since last time. However, I suggest to make another round of review to get a good paper out of the topic. Like it is now I would stil not recomment to publish it. Reasons:

line 28: why is this result very good? I would not say so. How is this result compared to other research? I miss this completely in the discussion.

where is the laser source? I asked this in my last review.

Most of the figures have a bad description or show unneccessary graphs.

E.g.

Figure 2 is still fuzzy

what is number of time in figure 19?

Results of Figure 21 should be expressed with one number e.g. Standard deviation, RMSE

Same with Figure 22 and Figure 23

I dont have time for more comments, but there are many more flaws like this in the paper.

but just for clarification, this is the state of the art:

https://www.youtube.com/watch?v=DUgjFaYyecE 

please refer to this

Author Response

Dear Editors:
Thank you for the opportunity to revise our manuscript, A Vision Servo System for Automated Harvest of Sweet pepper in Korean Greenhouse Environment. We appreciate the careful review and constructive suggestions. It is our belief that the manuscript is substantially improved after making the suggested edits. Following this letter are reviewer comments with our responses in italics, including how and where the text was modified. Changes made in the manuscript are marked using Red letters.
Thank you for your consideration.

Sincerely,

Round 3

Reviewer 3 Report

The paper is getting better now. However still not good enough:

Table 1 is not neccessary, refer information in text

Description of Table 2 is not self-explanatory

Figure 20: not neccessary, delete. Information lays in two numbers: mean and standard-deviation

Table 4: what is number of time? seconds? minuites?

Figure 21 number of time [no.]: I suggest you mean correction step? see also table 4

Description Figure 21 is not detailed enough

please use the same format (size, axis description, lining etc.) for all excel sheets! this looks so unprofessional when it is chaning for each figure

Table 5: you do not need Tables with one line. put Table 5-Table 7 in one

Talk in the discussion about the picking time, and discuss how many robots you need to replace one human.

Author Response

Dear Editors:

Thank you for the opportunity to revise our manuscript, A Vision Servo System for Automated Harvest of Sweet pepper in Korean Greenhouse Environment. We appreciate the careful review and constructive suggestions. It is our belief that the manuscript is substantially improved after making the suggested edits. Following this letter are reviewer comments with our responses in italics, including how and where the text was modified. Changes made in the manuscript are marked using Red letters.

Thank you for your consideration.

Sincerely,

The paper is getting better now. However still not good enough:

Table 1 is not neccessary, refer information in text

We agree with the reviewer and have deleted table 1. The description has been changed like line 310 to 314.

Description of Table 2 is not self-explanatory

We agree with the reviewer and have deleted table 2. The description has been changed like line 321 to 328.

Figure 20: not neccessary, delete. Information lays in two numbers: mean and standard-deviation

We agree with the reviewer and have deleted Figure 20. The description has been changed like line 335 to 339.

Table 4: what is number of time? seconds? minuites?

In order to satisfy the opinion of the reviewer, we have changed description like control count in table 2.

Figure 21 number of time [no.]: I suggest you mean correction step? see also table 4

Thank you for your advice. We agree with the reviewer and have changed like correction step in Figure 20.

Description Figure 21 is not detailed enough

We agree with the reviewer and have described like line 359 to 360.

please use the same format (size, axis description, lining etc.) for all excel sheets! this looks so unprofessional when it is chaning for each figure

Thank you for your advice. We agree with the reviewer and have changed unified format.

Table 5: you do not need Tables with one line. put Table 5-Table 7 in one

Thank you for your advice. We agree with the reviewer and have deleted table 5 and have put table 6 – table 7 in one.

Talk in the discussion about the picking time, and discuss how many robots you need to replace one human.

Thank you for your advice. In order to satisfy the opinion of the reviewer, we have described like line 443 to line 451.
